# Improving Out-of-Distribution Anomaly Detection with Domain-Invariant Latent Representations

## Abstract

Domain generalization focuses on leveraging the knowledge from the training data of multiple related domains to enhance inference on unseen in-distribution (IN) and out-of-distribution (OOD) domains. In our study, we introduce a multi-task representation learning technique that leverages the knowledge of multiple related domains to improve the detection of classes from unseen domains. Our method aims to cultivate a latent space from data spanning multiple domains, encompassing both source and cross-domains, to amplify generalization to OOD domains. Additionally, we attempt to disentangle the latent space by minimizing the mutual information between the input and the latent space, effectively de-correlating spurious correlations among the samples of a specific domain. Collectively, the joint optimization will facilitate domain-invariant feature learning. Using principles of domain generalization, we try to develop a robust anomaly detection model that can accurately identify anomalies even when those anomalies come from a distribution different from the training data. We assess the model's efficacy across multiple cybersecurity datasets, using standard classification metrics on both unseen IN and OOD sets, and validate the results with contemporary domain generalization methods.

## 1 Introduction

Domain generalization seeks to learn from multiple related source domains and subsequently evaluate the model on an unseen target domain Zhou et al. (2022); Wang et al. (2022). A domain typically constitutes a particular probability distribution from which the training samples are drawn. These samples stem from a source domain, while a separate target domain is kept for testing. Deep neural networks excel when tested on data from the same distribution as their training data. Within the few-shot learning (FSL) framework, models aim to adapt to out-of-distribution (OOD) data with minimal labeled samples Vuorio et al. (2019); Sung et al. (2018); Snell et al. (2017); Vinyals et al. (2016). Meta-learning endeavors to train models across varied tasks, enabling them to grasp new tasks with few samples Finn et al. (2017). A pivotal challenge arises from the domain shift between training and test distributions, compromising generalization Guo et al. (2020). Many recent studies train with both in-distribution and OOD data, though this can infuse bias. In the case of domain adaptation, it is considered that the information about the target domain or its classes is accessible during the training process Luo et al. (2019). Some approaches augment training data with perturbations mimicking the target domain. In domain generalization, the underlying assumption is that there's no foreknowledge of the target distribution. The challenge lies in designing a model that can classify data in unfamiliar domains once trained on single or multiple datasets without further training. We aspire to devise domain invariant features that bridge multiple related domains without directly engaging them during the training process. Previous research has proposed methods such as adversarial feature alignment using maximum mean discrepancy Muandet et al. (2013) to achieve domain generalization. In our case, we focus on learning a disentangled latent representation by consistently minimizing mutual information between prior and latent spaces from multiple related domains. However, our multi-domain latent space is curated with samples spanning multiple other domains(cross-domains) with the mutual invariance penalty trying to remove the spurious correlation present in individual domains. Recently, the work Hu et al. (2022) has highlighted the problem of how spurious correlation can hurt generalization performance when the task correlation

changes among the related tasks and solves the problem with multi-task causal representation learning with task-specific invariant regularization. In our case, we consider the scenario where the samples from multiple domains will have spurious feature correlations and therefore minimizing the mutual information content between the input and the latent space of their respective kernels will aim to remove the spurious correlation in the latent space. Otherwise, this is done by measuring the entropy or the uncertainty of the data on the eigenspectrum of the gram matrix of the input and latent space kernels. Multi-task learning facilitates learning representations from multiple diverse datasets and therefore jointly optimizing it with a de-correlation penalty to achieve generalization to unseen domains. In high-dimensional spaces, distances between points can become less meaningful due to the curse of dimensionality. Our motivation behind the encoder-decoder model or latent space analysis to detect unseen anomalies roots back to the fact that the principle of concentration of distance Beyer et al. (1999) reveals that for a query point p, its relative distance (or contrast) D to the farthest point and the nearest point converges to 0 with the increase of dimensionality d. This means that the discriminative power between the nearest and the farthest neighbor becomes poor in a high-dimensional space. Consequently, using latent space embeddings rather than high-dimensional features is a preferred approach for anomaly detection.

In summary, our contributions can be highlighted as:

- We consider the scenario where each domain has its own spuriously correlated features which hurts the generalization performance of the model when tested on OOD domains. To address this, we propose a multi-task representation learning approach that adapts a multi-domain latent space by incorporating data from multiple sources and cross-domains. Simultaneously, we try to cultivate a disentangled or invariant latent space by applying a regularization penalty that minimizes the mutual information(MI) between the input and the latent space.

- Our methodology jointly optimizes the classification loss, the multi-domain reconstruction loss, and the mutual invariance regularization in the latent space. The regularization, calculated in the kernel space of the input and latent space, acts as a de-correlation penalty, or otherwise, we do not want the spurious correlation of the higher dimensional features to be projected to the latent space.

- We show that cross-domain data when added in a principled way, can improve generalization performance on the IN and OOD classes. In our case, incorporating data with varying correlation structures effectively introduces spurious correlation during training, which serves as a straightforward and natural method to bias the network toward learning invariant representations We perform a wide range of experiments on a number of i.i.d datasets from IoT, cybersecurity, healthcare, and one time series(non-i.i.d) dataset to showcase overall improvement in generalization performance across IN, OOD, and cross-domain classes, most importantly the OOD domains.

## 2 Related Work

Domain generalization techniques can be grouped into the following primary categories: domain invariant representation learning, meta-learning,latent dimension regularization, and metric learning.

*1) Domain Invariant Representation Learning:* This method aims to identify domain invariant representations that can be extended to unseen domains. The crux of these strategies, as seen in works such as Seo et al. (2020), is to filter out domain-specific insights while maintaining cross-domain information. Notable studies employing autoencoders, such as Ghifary et al. (2015), amalgamate multiple domains during training, augmented by data enhancement techniques, to extract domain-invariant characteristics. These features then demonstrate superior generalization to out-of-distribution data. Another study, Maximum Mean Discrepancy Adversarial Autoencoder (MMD-AAE) Li et al. (2018b), in the context of few-shot learning, emphasizes aligning varied domain distributions to a generic prior distribution while engaging in adversarial feature learning. An innovative approach is suggested in Chattopadhyay et al. (2020), where a domain-centric masking technique is applied to learn both domain-specific and domain-invariant features. This will facilitate efficient source domain classification and sufficient generalization to target domains. In Liang et al. (2021), a noise-enhanced supervised autoencoder is presented. This autoencoder undertakes a dual task of reconstructing inputs and classifying both the inputs and the reconstructed entities by feeding them back as input

to the model. Leveraging the intra-class correlation (ICC) metric, they demonstrate the superior discriminative and generalizing capabilities of the learned feature embeddings. Moreover, authors in Jin et al. (2020) propose domain generalization through domain-invariant representation that uniformly distributes across multiple source domains. Their approach employs moment alignment of distributions and enforces feature disentanglement via an entropy loss. Recently, the authors Lu et al. (2022) of the DIFEX paper utilized a knowledge distillation framework to capture the high-level Fourier phase as internally-invariant characteristics, while simultaneously learning cross-domain correlation alignment to extract mutually-invariant features. Zhao et al. (2019) proposed an upper and lower bounds generalization to address the class conditional shift or the fact that the class conditional distribution of input features changes between source and target domains for domain adaptation. Previously, the problem of domain generalization has been addressed using regularization techniques. The authors Zhao et al. (2019) formulate the problem of finding the right regularizer as a meta-learning problem and show that the learned regularizers have good cross-domain generalization ability.

*2) Meta-learning:* This approach employs learning from a number of related tasks for domain generalization, as observed in works such as Finn et al. (2017); Li et al. (2018c;a). The study in Erfani et al. (2016) introduces a technique to discern a domain interdependent projection leading to a latent space. This space minimizes biases in the data while preserving the inherent relationship across multiple domains. Model Agnostic Meta-Learning (MAML) has also been extended to latent dimension settings by performing the gradient-based adaptation in the low dimensional space instead of the higher dimensional space of model parameters Rusu et al. (2018). Zero-shot learning Wang et al. (2019) aims at learning models from seen classes and inferring on samples whose categories were unseen during the training process.

*3) Information Bottleneck Principle and Metric Learning:* In contrast to the aforementioned methodologies, our strategy propels direct disentanglement or decorrelation between multiple training domains. An information-theoretic perspective on variance-invariance-covariance has been provided here Shwartz-Ziv et al. (2023) in the context of self-supervised learning which helps to achieve generalization guarantees for downstream supervised learning tasks. Adversarial learning-based domain adaptation methods are prone to negative transfer which hurts the generalization performance Jeon et al. (2021). In our method, class-invariant features are indirectly learned in the latent dimension by minimizing mutual information between latent and prior space. Metric learning aims to learn a representation function that can map higher-dimensional data to a latent embedded space. The distance in embedded space should preserve the similarity between the similar classes and the dissimilar classes are pushed away from each other. The authors Venkataramanan et al. (2021) propose mixing target labels with training samples to improve the quality of representations or embeddings for classification purposes. In deep metric learning, we try to learn non-linear mapping from input space to low-dimensional latent space Oh Song et al. (2016). Various mix-up techniques have been recently proposed to improve the quality of representation learned by adopting some interpolation techniques between pairs of input samples and their labels Zheng et al. (2019).

*4) Other Related Works:* The authors Hendrycks & Gimpel (2016) suggest using the statistics of softmax outputs to estimate both the probability of error and the likelihood of a test sample being out-of-domain. They compare the performance of this approach by directly using the raw softmax output probabilities as a measure of confidence. The paper Xu et al. (2018) addresses the problem of domain shift when a learned model tends to degrade heavily on a target domain via unsupervised domain adaptation by learning a common feature map from multiple source domains by minimizing the domain distribution discrepancy between those multiple source domains. This work Yang et al. (2022) acknowledges the fact that Out-of-distribution (OOD) detection is critical for safety-critical applications but lacks a unified benchmark, leading to inconsistent comparisons, prompting the development of the OpenOOD codebase to provide a comprehensive benchmark. They also emphasize how OOD detection is closely linked to related fields such as anomaly detection (AD), open set recognition (OSR), and model uncertainty, as methods created for one area are often applicable to others. The authors Sun et al. (2022) explore the use of non-parametric nearest-neighbor distance for out-of-distribution detection, offering greater flexibility and demonstrating superior performance without relying on strict distributional assumptions. One significant work in this direction Arjovsky et al. (2019) which highlights the fact that just minimizing the training error ignores the implicit data biases and recklessly absorbs all correlations found in training data which hurts the generalization

performance. The authors develop causal tools to develop methods for distinguishing spurious and invariant correlations, reducing machine learning systems' dependence on data biases and improving their ability to generalize to new test distributions.

On the other hand, our idea is based on the principle of invariance and disentanglement Achille & Soatto (2018) which states that invariance to irrelevant factors in a deep neural network corresponds to minimizing the information content of the learned representation. Adding noise during training naturally steers the network toward learning invariant representations. In our case, the cross-domain data with different kinds of correlation structures helps add spurious signals. We follow a multi-task representation learning approach that aims to learn a robust feature representation in the latent space by leveraging multiple sources and cross-domain feature information while simultaneously de-correlating the spurious feature correlation information of the input space. Following the sufficiency and minimality conditions of the information bottleneck principle, we try to create a latent space representation that contains only relevant information about the output tasks and discard the class-specific spurious correlation information of the input samples.

## 3 Problem Formulation

In our domain generalization problem, we consider $M$ training tasks comprising of source and cross-domain data consisting of $N$ samples, $\mathcal{S}_{train} = \{X^i, Y^i | i = 1, 2, 3..., M\}$ where, $X^i, Y^i = \{(x_j^i, y_j^i)\}_{j=1}^{N_i}$. Let $I = \{I_1, I_2, ..., I_k\}$ represent the power set of these labels, where $I_q \subseteq \{0, 1, ..., K\}$. For example, if $I_1$ is a source domain, where, $I_1 = \{0, 1, 2, 3, 4\}$, then, we add multiple cross-domain data $I_2, I_3, I_4$, where, $I_2 = \{5, 6, 7\}$, $I_3 = \{8, 9, 10\}$, and $I_4 = \{11, 12, 13\}$. The extension to multiple domains necessitates the definition of a multi-task learning objective over all the M source and cross domains which can be given as

$$\mathcal{L}_{rec}(S_{train}; \theta, \phi) = \sum_{i=1}^{M} \left\| f_\theta^{(i)}\left(g_\phi^{(i)}\left(X^i\right)\right) - X^i \right\|_2^2 \tag{1}$$

In this expression, $g_\phi^{(i)}$ and $f_\theta^{(i)}$ denote the encoder and decoder functions respectively for each of the $M$ sources and cross domains, $X^i$ is the input training data from a particular domain. The total reconstruction loss is the sum of reconstruction losses over the source and cross-domain latent space reconstructions. The cross-domain data aims to introduce more diversity in the latent space representation which in turn can help to achieve better generalization performance.

### 3.1 Mutual Invariance Regularization

In order to learn a domain-invariant latent space, we use a kernel-based mutual information minimization between the prior and the latent space as a regularization technique.

#### 3.1.1 Kernel-based Renyi's Entropy and Joint Entropy to measure Mutual Information between Input and Latent Space

In information theory, the dependence measure or the total correlation between the feature variables is measured as the statistical independence in each dimension and is expressed as the Kullback Leibler(KL) divergence between the joint probability distribution and the marginal distribution of the features Yu et al. (2021). In our case, we enforce the de-correlation between the input and the latent kernel space comprising multiple source and cross-domains by minimizing the mutual information between them as a regularization penalty. This will try to introduce a disentangled latent space that is devoid of the spurious correlations in the individual domains in their high-dimensional input space.

The matrix-based Renyi's second-order entropy Yu et al. (2021) of a normalized positive definite(NPD) matrix $\mathcal{K}_x$, of size $l \times l$ in the input space, where $l$ represents a batch size, can be estimated as

$$\hat{H}_2(\mathcal{K}_x) = \frac{1}{1-\alpha} \log_2 \left( \sum_{i=1}^{l} \lambda_i(\mathcal{K}_x)^\alpha \right), \tag{2}$$

where the Gram matrix $\mathcal{K}_x$ is obtained by evaluating the positive definite (PSD) kernel on all $l$ pairs of samples in a batch of training, that is, $(\mathcal{K}_x)_{i,j} = \mathcal{K}_x(x_i, x_j)$ and $\lambda_i(X)$ denotes the $i^{th}$ eigenvalue of the input kernel matrix $\mathcal{K}_x$ of the $l_{th}$ batch, $\mathcal{K}_{i,j} = \frac{1}{l}\frac{\mathcal{K}_{i,j}}{\sqrt{\mathcal{K}_{i,i}\mathcal{K}_{j,j}}}$ Here, $\alpha = 2$ considering Renyi's second-order entropy.

Similarly, Renyi's quadratic entropy of the latent space kernel $\mathcal{K}_Z$ of size $l \times l$ is estimated as

$$\hat{H}_2(\mathcal{K}_z) = \frac{1}{1-\alpha} \log_2 \left( \sum_{i=1}^{l} \lambda_i(\mathcal{K}_z)^\alpha \right), \tag{3}$$

The argument in equation (3) is called the information potential. In the above section, we use the matrix-based second-order Renyi's entropy ($\alpha = 2$) Yu et al. (2021) to evaluate the entropy or the uncertainty of the latent and the input space in terms of the normalized eigenspectrum of the Hermitian matrix of the projected data in the Hilbert space. Now, we can estimate the matrix-based second-order joint entropy between latent space kernel $Z$ and the input space kernel $X$ as

$$\hat{H}_2(\mathcal{K}_X, \mathcal{K}_Z) = H_2\left(\frac{\mathcal{K}_x \circ \mathcal{K}_Z}{tr(\mathcal{K}_X \circ \mathcal{K}_z)}\right), \tag{4}$$

where $\circ$ represents the Hadamard product. Based on the above definitions, we calculate the joint entropy of the latent and the input space with the help of the matrix-based normalized Renyi's entropy of the latent space and the input space kernels. The joint entropy is used to derive the mutual information between the input and the latent space.

### 3.1.2 The Mutual Information Divergence

We use the matrix-based mutual information divergence to estimate the mutual information between the latent and input space kernels. Minimizing the mutual information indirectly results in de-correlating the feature correlation that exists in the original input space which helps in improving the generalization performance. The mutual information during each batch of the training can be estimated as

$$\hat{MI}(\mathcal{K}_X; \mathcal{K}_Z) = \hat{H}_2(\mathcal{K}_X) + \hat{H}_2(\mathcal{K}_Z) - \hat{H}_2(\mathcal{K}_X, \mathcal{K}_Z), \tag{5}$$

where $\hat{H}_2(\mathcal{K}_X, \mathcal{K}_Z)$, is the second-order joint entropy between the latent and the input kernel space. Minimizing this divergence as a regularization penalty in the final loss objective will aid in preserving useful disentangled information in the latent space during each iteration of the training process.

### 3.2 The Multi-Task Learning Objective

In our latent space multi-task learning approach, we leverage the label information of the multiple source and cross-domain encoded data in the latent space during the training process. In our approach, we do a joint optimization of the classification and the reconstruction loss along with the MI regularization in the latent space. The total loss calculated over all the $N$ batches, each consisting of $l$ samples, during the training process can be given as

$$\mathcal{L}(\mathcal{S}_{train}, Z; \phi, \theta) = \min_{\phi, \theta} \frac{1}{N} \sum_{l=1}^{N} \Big\{ \mathcal{L}_{ce}\left(g_\phi\left(X^l\right), y^l\right)$$
$$+ \beta \cdot \mathcal{L}_{MI}\left(X^l; Z^l\right) + \lambda \cdot \mathcal{L}_{rec}\left(X^l; \phi, \theta\right) \Big\}, \tag{6}$$

where, $\mathcal{L}_{ce}$ is the cross-entropy loss calculated on the latent space encoding considering the binary classification problem, given as,

$$\mathcal{L}_{ce}\left(g_\phi\left(X\right), y\right) = -\left(y \log\left(\mathcal{S}_y\left(g_\phi\left(X\right)\right)\right) + (1-y) \log\left(1 - \left(\mathcal{S}_y\left(g_\phi\left(X\right)\right)\right)\right)\right)$$

$\mathcal{L}_{MI}$ is the disentanglement or de-correlation loss between the latent space and the input space expressed in the form of mutual information divergence measured in their kernel space, given in eq: 5, $\mathcal{S}_y$ is the softmax

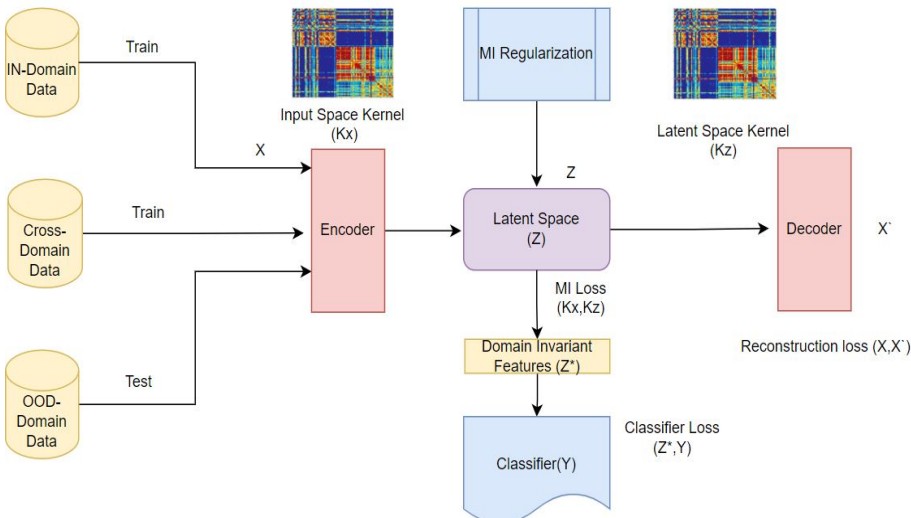

Figure 1: The Multi-task Latent Space Regularized Encoder-Decoder Model (MTLS-RED).

function applied on the encoded data $g_\phi(x)$, $\mathcal{L}_{rec}$ is the reconstruction loss, $\phi$, $\theta$ are the encoder and decoder parameters. Here, we augment the cross-entropy loss with the mutual information regularization between the latent and prior space. The regularization aims to penalize spurious information content between the latent encoding and the input space, the strength of which is determined by the hyper-parameter $\beta$. $\beta$ controls the trade-off between decorrelation and classification. During the joint optimization, we try to balance the reconstruction weight and the mutual information regularization.

## 4 Experiments

In this section, we demonstrate the performance of our proposed model on benchmark IOT datasets.

### 4.1 Datasets

- **CSE-CIC-IDS2018** This is a publicly available cybersecurity dataset that is made available by the Canadian Cybersecurity Institute (CIC). It consists of 7 major kinds of intrusion datasets. We use SOLARIS, GOLDENEYE as source domain data, INFILTRATION, BOTNET as cross-domain data, and RARE, SLOWHTTPS, HOIC as OOD data, along with a OOD BENIGN class from a different day.

- **CICIoT 2023** This is a state-of-the-art dataset for profiling, behavioral analysis, and vulnerability testing of different IoT devices with different protocols from the network traffic, consisting of 7 major attack classes. We use BENIGN, DoS, and DDoS as source data, RECON, as cross-domain data and WEB, MIRAI as OOD test data.

- **CICIoMT 2024** This is a benchmark dataset to enable the development and evaluation of Internet of Medical Things (IoMT) security solutions. The attacks are categorized into five classes. We use BENIGN, DDoS, DoS as source-domain, RECON, and SPOOFING as cross-domain, and MQTT as OOD data.

- **Arrythmia** This dataset is about atrial fibrillation (also called AFib or AF) which is a quivering or irregular heartbeat (arrhythmia) that can lead to blood clots, stroke, heart failure, and other heart-related complications. The dataset contains five classes/categories: N (Normal), S (Supraventricular ectopic beat), V (Ventricular ectopic beat), F (Fusion beat), and Q (Unknown beat).

- **EMG Gesture Recognition Dataset** The EMG Signal for Gesture Recognition dataset is a widely used dataset in the field of bioinformatics and human-computer interaction, particularly for

developing systems that recognize hand gestures based on electromyography (EMG) signals. The dataset consists of EMG signals collected using sensors placed on the forearm. The dataset typically includes recordings for multiple hand gestures. Common gestures include fist, open hand, pointing, and various finger movements.

## 4.2 Baselines

We consider the following models related to latent space multi-task representation learning and few-shot learning as baselines to compare and evaluate the performance of our proposed model. More details about the baselines can be found in the supplementary material.

- **Correlation Alignment for Deep Domain Adaptation (CORAL)** Sun & Saenko (2016) This paper adopts an unsupervised domain adaptation technique that tries to align the covariance of the source and the target distributions. We extend CORAL to latent space and align the source and cross-domain data correlation to improve generalization on OOD data.

- **Deep Multi-task Autoencoder (MTAE)** Ghifary et al. (2015) In this latent dimension encoder-decoder-based model the reconstruction error is optimized over multiple domains in a supervised fashion. We consider multiple sources and cross-domain data and train them jointly with the label information in latent space. This is basically our model without the de-correlation regularization.

- **Minimum Mean Discrepancy-Autoencoder(MMD-AE)** Li et al. (2018b); Sathya et al. (2022) This paper uses the MMD measure as regularization for domain generalization between multiple cross-domain data. We use it as a few-shot learning method where the cross-domain data are added to improve the OOD generalization.

- **Noise Enhanced Supervised Autoencoder (NSAE)** Liang et al. (2021)This model jointly predicts the labels of inputs and reconstructs the inputs considers reconstructed samples as noisy inputs and feeds them back to the model. In this model, requires doing an extra step of fine-tuning by further training the reconstructed data using a supervised classifier.

- **DIFEX** Lu et al. (2022) This recent paper uses the concept of mutual invariance to learn features from multiple cross-domains and use this information to classify OOD domains. This authors argue that domain-invariant features should come from both internal and mutual invariance, with internal invariance capturing intrinsic semantics within a single domain, and mutual invariance capturing common information across domains.

## 4.3 Training Strategy

In the training phase, we jointly optimize multiple tasks by considering multiple source and cross-domain data in the latent space. Our goal is to jointly train the latent space with datasets that have different patterns of correlation structure among its features which can be considered as equivalent to adding more noise to the data in the input space. Minimizing this noise will cultivate domain-invariant latent space for better domain generalization. For example, if two or more features of any two attack datasets exhibit distinct correlation patterns, we add one as the source domain and the other as cross-domain data. This will introduce more diversity in the input space as we are now considering data from different domains with non-overlapping correlation patterns. By applying the mutual information penalty in the kernel spaces of the input and the latent, we attempt to create a disentangled latent space which aims to classify the OOD classes. Basically, we allow the model to learn a diverse latent space representation by reconstructing different source and cross-domain data while simultaneously de-correlating domain-specific spurious correlation information. During training, we vary the cross-domain data in various proportions ranging from 0%-40% of the source domain samples and try to understand the efficacy of our multi-domain disentangled latent space in evaluating the classification performance on the out-of-distribution(OOD) classes. In all our experiments, we keep the rare anomaly classes, that is, the ones with the least number of samples available as the OOD class.

### 4.3.1 Selecting the cross-domains, source and OOD domains

We consider a setting where the joint distribution of the data doesn't have much overlap but the marginal distribution of the features does have an overlap Dong & Ma (2022). This happens in our case when the features exhibit different correlation patterns between the source and cross-domain data distributions. For example, in Fig 2, we observe that DOS and DDOS datasets have similar correlations between the features. However, the correlation between the features in the other attacks like MIRAI and WEB is different. Similarly, the GOLDEN and SOLARIS datasets have a certain kind of correlation between the features that differ from other attack datasets like INFIL and BOTNET. The two datasets must have overlapping marginal distribution which is required for extrapolation. Our goal is to combine the datasets with different patterns of correlation structure among its features during the training phase in order to increase the diversity of the input space.

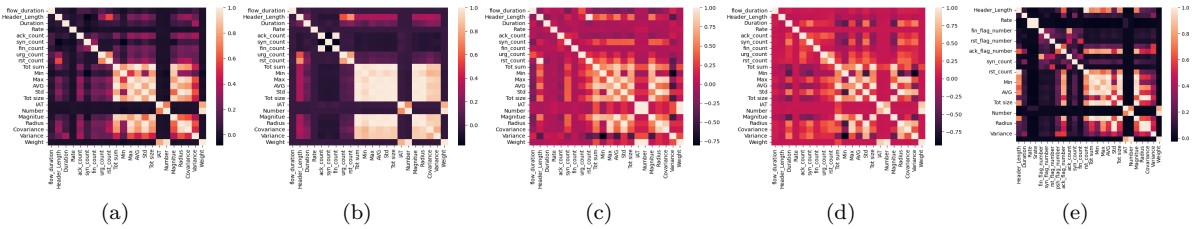

|     |     |     |     |     |
|-----|-----|-----|-----|-----|
| (a) | (b) | (c) | (d) | (e) |

Figure 2: Fig (a-o) Correlation plots of various cyber-security datasets. (a)DOS-IOT (b) DDOS IOT (c) WEB (d) MIRAI (e) BENIGN IOT of CIC-IDS DATASET

[1]

---

**Algorithm 1** The Multi-task Latent Space Regularized Encoder-Decoder Model (MTLS-RED)

---

Input: Source domain data $X_1, X_2, ..., X_m$, $X \in \mathcal{R}^d$, $\forall m \in \{1, 2, 3, ..\}$,
Add cross-domain data $X'_1, X'_2, ..., X'_m$, $X' \in \mathcal{R}^d$, $\forall m \in \{4, 5, 6, .., \}$,
Let the source and cross-domain labels be $\{y_i^m\}_{i=1}^n$, $\forall m \in \{1, 2, 3, 4, 5, 6, ..., M\}$
Initialize the encoder(E) and decoder(D) weights: $\mathbf{W}_\phi \in \mathcal{R}^{d_x \times d_z}$, $\mathbf{W}_\theta \in \mathcal{R}^{d_z \times d_x}$ , learning rate $\alpha$
**while** not end of epochs **do**:
    **for** batch = 1 to total batches N **do**:
        Sample mini-batch data $\{X_i\}_1^l \in \mathbf{R}^d$, $l$ is batch-size
        Compute the RBF kernels of input space $\mathcal{K}_{x_l}$ and latent space $\mathcal{K}_{z_l}$ of size $l \times l$
        Calculate mutual information between input space $X_l$ and latent space $Z_l$ using matrix-based Renyi's entropy $MI\left(\mathcal{K}_{X_l}; \mathcal{K}_{Z_l}\right)$.
        Perform a forward pass on encoder $E\left(X_{\phi_i}\right)$
        Calculate total batch loss $\mathcal{L}_l = \mathcal{L}_{ce}\left(X^l, y^l\right) + \mathcal{L}_{rec}\left(X^l, X^{l'}\right) + \lambda \mathcal{L}_{\mathcal{MI}}\left(X^l \| Z^l\right)$
        Update $\mathbf{W}_\phi$ and $\mathbf{W}_\theta$
        $\mathbf{W}_{\phi_{t+1}} \leftarrow \mathbf{W}_{\phi_t} - \alpha_1 \nabla_\phi \mathcal{L}_l\left(\theta, \phi\right)$
        $\mathbf{W}_{\theta_{t+1}} \leftarrow \mathbf{W}_{\theta_t} - \alpha_2 \nabla_\theta \mathcal{L}_l\left(\theta, \phi\right)$
    **end for**
**end while**
Output: Trained MTLS-RED algorithm

---

### 4.4 Hyperparameter Sensitivity

In the joint optimization objective, we aim to balance the weightage to the hyperparameters $\beta$ and $\lambda$ to achieve good generalization performance to all the classes. We find that allocating a higher weightage to the entropy loss and keeping less weightage to the reconstruction loss results in the best overall performance of

---

[1]https://anonymous.4open.science/r/MTRAE-712B/mtrae/multitaskAE_orig.py

| Datasets | Features | Sig Feats | learning rate | kernel bw | batch size | Weightage | Latent Dim |
|---|---|---|---|---|---|---|---|
| CSE-CIC-IDS | 79 | 30 | 0.005 | 0.01-6 | 200 | $\lambda, \beta = 0.6, 2$ | $79 - 30 - 20 - 13$ |
| CICIoT 2022 | 43 | 21 | 0.005 | 0.01-6 | 200 | $\lambda, \beta = 0.6, 2$ | $43 - 21 - 15 - 11$ |
| CICIoMT 2024 | 44 | 25 | 0.005 | 0.01-6 | 200 | $\lambda, \beta = 0.6, 2$ | $44 - 25 - 18 - 11$ |
| Arrythmia | 34 | 20 | 0.005 | 0.01-6 | 200 | $\lambda, \beta = 0.6, 2$ | $34 - 20 - 13 - 8$ |
| EMG | 8 | 8 | 0.005 | 0.01-6 | 400 | $\lambda, \beta = 0.6, 2$ | $8 - 5 - 3$ |

the model considering different scenarios. The kernel bandwidth used to estimate the matrix-based Renyi's quadratic entropy in the latent space and input space proves to be a significant hyperparameter. Optimum bandwidth selection is directly linked to the smoothness of the learning. A small bandwidth can lead to under-smoothing whereas a large bandwidth may result in over-smoothing We vary the kernel bandwidth during our experiments. We consider the features which has maximum correlation with the label in the case of each dataset. For a fair comparison between the baselines, we keep the latent space dimension similar while experimenting on the datasets with a particular baseline. In Fig 3, we perform some experiments to find the optimal value of the $\lambda$ and $\beta$ parameters. We observe that in most cases maximum accuracy is obtained when the MI penalty ($\beta$) is between 1 and 2 with most of the IN and OOD datasets. We keep the reconstruction loss weightage($\lambda$) fixed while conducting those experiments.

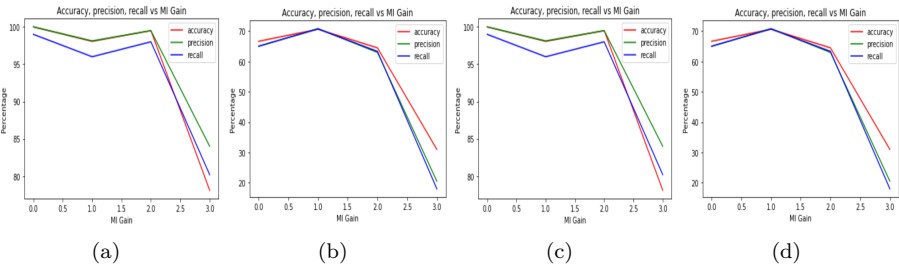

| (a) | (b) | (c) | (d) |
|---|---|---|---|

Figure 3: Fig shows MI weightage ($\beta$) variation with classification metrics on SOURCE and CROSS DOMAIN data (SOLARIS, GOLDENEYE, BENIGN,BRUT-FORCE, INFIL, BOTNET) and OOD DATA(RARE, HOIC, SLOW-HTTPS, BENIGN). We keep reconstruction loss weightage fixed. Fig shows that maximum accuracy is obtained when MI is between 1 and 2 for all the cases.

## 4.5 Ablation Study

We perform different kinds of ablation studies to validate the performance of our model and also to compare our proposed model with standard baselines. In our experiments, we consider multiple source and cross-domain reconstructions and use the encoded space to classify source, cross-domain, and OOD data. Our setup is different from MTAE Ghifary et al. (2015) where the authors adopt a two-stage approach where the model is trained first, and then the trained encoder is used for classification purposes. On the other hand, our strategy is to jointly optimize all the components together to improve the generalization performance. The joint objective promotes discriminative power while simultaneously reducing the dependency on spurious data-specific correlation. We try to design a principled approach to how cross-domain data can be combined with particular source domain data in order to improve the generalization to out-of-distribution data samples. We vary the cross-domain data in various proportions 0%-40% of the source data in the training data in order to understand the impact of the de-correlation penalty under different scenarios. In Table 1,2, we observe that as we gradually increase the percentage of the cross-domain classes in the training data, the performance of most of the baseline models deteriorates on the IN distribution. Whereas, with a small percentage of cross-domain data initially, most of the models overfit to the IN distribution datasets. The proposed model, on the other hand, gives an overall good performance on all the source, cross-domain, and OOD classes as we vary the amount of cross-domain data. It is to be noted that the OOD-domain data are never observed during any stage of the training process. In Table 1, we observe an interesting phenomenon

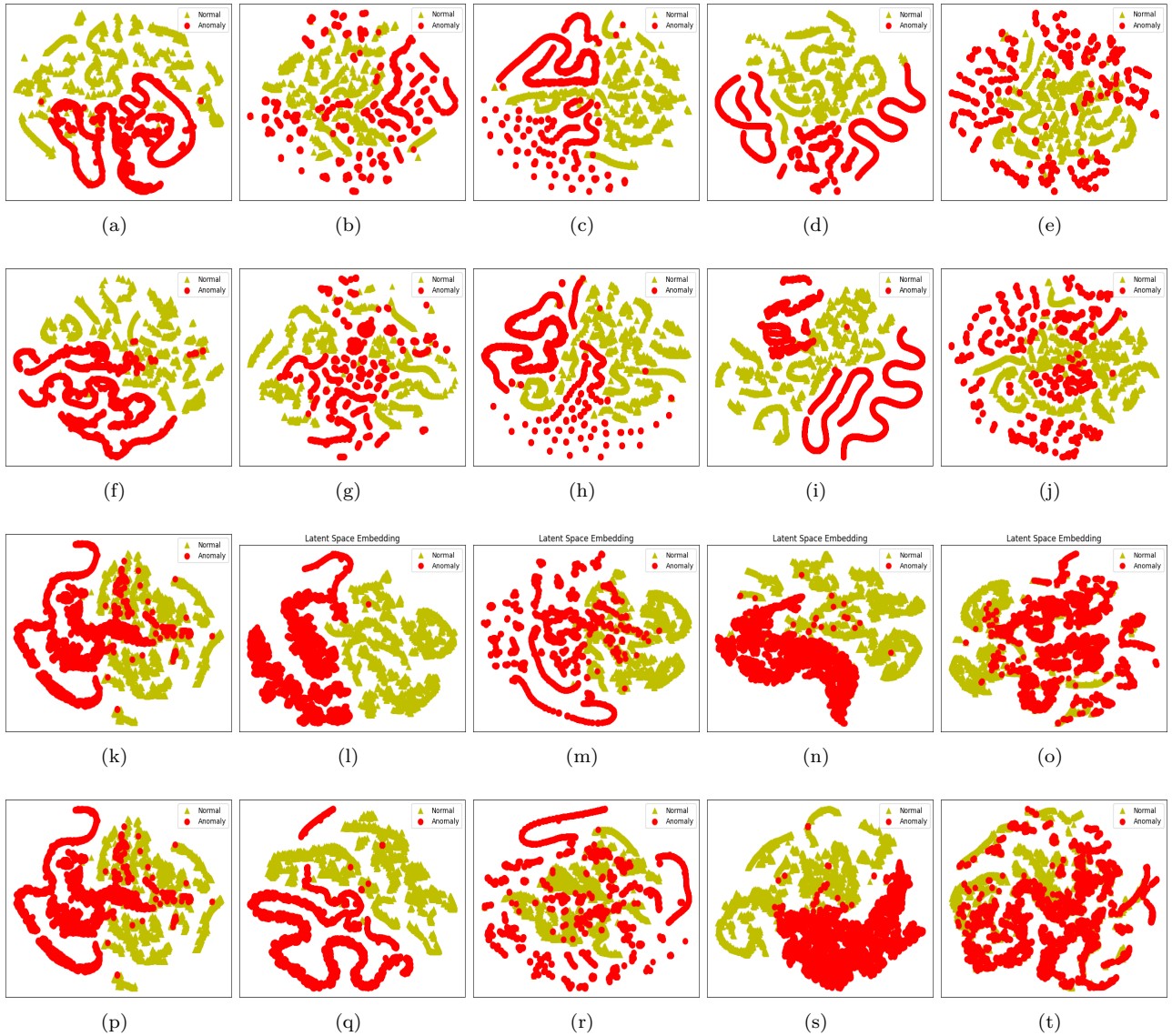

Figure 4: Fig (f-j),(p-t) T-SNE projection of the latent space of DMTAE (without MI regularization) and (a-e),(k-o) is our proposed MTL-RED (with MI regularization). Again, (a-j) the attacks are GOLDEN-EYE, SOLARIS, BOTNET, HOIC, RARE. (k-t) the attacks are DOS, DDOS, MQTT, RECON, and SPOOFING. We see the difference distinctively in SOLARIS(b,g), RARE(e,j), DOS(k,p), DDOS(l,q) and MQTT(m,r).

| Model | Percentage | SOURCE DOMAINS | | CROSS DOMAINS | | | OOD DOMAINS | | |
|---|---|---|---|---|---|---|---|---|---|
| | | SOLARIS | GOLD-EYE | INFIL | BOTNET | RARE | HOIC | HTTPS | BENIGN |
| DMTAE | 0% | **99.97** | 52.70 | 04.00 | 09.50 | 00.30 | 00.38 | 02.30 | 97.98 |
| | 15% | 99.50 | **99.70** | **99.40** | **99.50** | 61.30 | 27.38 | 32.30 | **98.11** |
| | 25% | 81.60 | 79.60 | 69.70 | 78.50 | 65.40 | **50.00** | 42.40 | 62.00 |
| | 40% | 61.61 | 61.30 | 31.30 | 61.60 | **72.60** | 48.50 | **69.80** | 83.32 |
| MMD-AE | 0% | **99.99** | **99.29** | 00.53 | **99.98** | 55.47 | 99.98 | 99.69 | 99.71 |
| | 15% | 99.98 | 02.55 | **22.55** | 99.83 | 12.19 | 41.34 | 98.77 | 98.67 |
| | 25% | 99.78 | 02.52 | 15.35 | 99.83 | 12.19 | 41.27 | 99.69 | 98.61 |
| | 40% | 99.96 | 01.88 | 12.30 | 99.83 | 14.13 | 41.12 | 56.21 | 99.14 |
| NSAE | 0% | 99.99 | 99.98 | 00.09 | 99.99 | **77.03** | 99.90 | **97.02** | 99.58 |
| | 15% | 99.80 | **99.99** | 04.82 | 34.92 | 36.04 | 00.00 | 05.16 | **99.80** |
| | 25% | **99.99** | 03.16 | 00.32 | **99.83** | 59.36 | 15.90 | 00.35 | 99.69 |
| | 40% | 99.98 | 34.36 | **53.66** | 99.83 | 12.36 | 00.00 | 19.33 | 99.40 |
| CORAL | 0% | 59.18 | 90.61 | 30.45 | 00.80 | 08.40 | 55.06 | 03.40 | 69.94 |
| | 15% | 61.53 | 12.64 | **31.79** | 50.43 | **33.74** | 41.31 | 50.38 | 67.54 |
| | 25% | 38.85 | **99.99** | 0.00 | 0.01 | 22.96 | 82.21 | 0.00 | 95.01 |
| | 40% | **99.99** | 38.85 | 00.01 | 00.00 | 22.96 | **82.21** | 00.01 | **99.19** |
| MTLS-RED | 0% | **88.83** | **86.08** | **82.41** | **96.12** | 28.01 | 80.99 | 73.93 | 73.23 |
| | 15% | 79.00 | 78.85 | 78.71 | 78.68 | 78.88 | 79.13 | 78.82 | 79.01 |
| | 25% | 83.90 | 80.70 | 80.70 | 83.80 | 76.50 | **81.96** | **85.00** | 77.10 |
| | 40% | 79.46 | 79.33 | 79.20 | 79.12 | **78.99** | 79.33 | 79.25 | **79.64** |

Table 1: Performance (accuracy) of the proposed and baseline methods with different percentages of cross-domain data added to the source domain during training. The results are on the CIC-CSE-IDS dataset. We highlight the best performance of each model across all the datasets. The results are averaged over 4-5 iterations of training for each case.

| Model | Percentage | SOURCE DOMAINS | | CROSS DOMAINS | | | OOD DOMAINS | | |
|---|---|---|---|---|---|---|---|---|---|
| | | DDOS | DOS | RECON | SPOOF | MQTT | MIRAI | WEB | BENIGN |
| DMTAE | 0% | 99.96 | 99.99 | 54.07 | 46.98 | 78.43 | **99.97** | 30.55 | 97.53 |
| | 15% | 99.99 | 99.99 | 98.66 | 71.84 | 78.43 | 79.38 | **30.55** | 97.53 |
| | 25% | 99.99 | 99.99 | **99.99** | **75.55** | **99.93** | 80.07 | 21.38 | 98.63 |
| | 40% | **99.99** | **99.99** | 98.534 | 74.43 | 89.42 | 80.83 | 29.02 | 97.78 |
| MMD-AE | 0% | 99.96 | 99.96 | 49.95 | 41.32 | **95.87** | 84.01 | 21.15 | **98.56** |
| | 15% | **99.99** | **99.99** | 98.47 | 70.54 | 90.54 | **84.01** | 42.53 | 96.49 |
| | 25% | 99.99 | 99.99 | 99.11 | 73.60 | 89.42 | 77.60 | 58.04 | 93.48 |
| | 40% | 99.61 | 99.46 | **99.30** | **78.23** | 94.84 | 72.08 | **67.21** | 92.41 |
| NSAE | 0% | **99.99** | 99.99 | 46.90 | 32.42 | 69.11 | **99.90** | 56.05 | 94.68 |
| | 15% | 99.99 | **99.99** | 97.77 | 69.11 | **99.90** | 99.87 | 36.66 | **97.71** |
| | 25% | 99.99 | 99.99 | **98.70** | **71.29** | 99.71 | 99.90 | 43.83 | 96.11 |
| | 40% | 99.99 | 99.99 | 98.62 | 70.64 | 99.89 | 99.90 | 49.70 | 95.78 |
| CORAL | 0% | **99.99** | 99.99 | 98.53 | 74.43 | 89.42 | **81.13** | 30.53 | 98.73 |
| | 15% | 99.99 | **99.99** | 98.53 | 74.43 | 89.42 | 81.13 | **42.30** | 98.73 |
| | 25% | 99.99 | 99.99 | **98.86** | 75.55 | **99.90** | 80.08 | 42.03 | **98.73** |
| | 40% | 99.99 | 99.99 | 99.11 | **77.21** | 99.57 | 80.51 | 42.30 | 98.73 |
| MTLS-RED | 0% | 99.99 | 99.99 | 98.41 | 78.92 | 78.21 | 76.04 | 73.91 | 87.67 |
| | 15% | 99.99 | 99.99 | 98.41 | 78.92 | 78.21 | 99.75 | 68.03 | 91.83 |
| | 25% | 99.99 | 99.99 | 98.41 | 78.21 | 53.19 | 99.87 | 73.67 | 93.05 |
| | 40% | **99.99** | **99.99** | **98.98** | **81.17** | **95.87** | **99.88** | **75.91** | **91.2** |

Table 2: Performance (accuracy) of the proposed and baseline method with different percentages of cross-domain data added to the source domain during training. The results are on the combined features of CIC-IOT and CIC-IoMT datasets. The results are averaged over 4-5 iterations of training for each case.

where we gradually increase the cross-domain data to $30\% - 40\%$, by adjusting the kernel bandwidth and the hyperparameters $\beta$ and $\lambda$, we can train a model that shows an overall generalization performance across all training and test datasets. The base models mostly overfit to the training data or a particular cross-domain data. In Table 2, we find that when we increase the proportion of cross-domain data during the training, the classification results are significantly good across the OOD datasets like WEB and SPOOFING attacks. In Table 3, we showcase the performance of our method with the Arrhythmia dataset where we train the model

| Model | Anomaly(%) | SOURCE DOMAINS | | CROSS DOMAINS | | OOD DOMAINS |
| | | VEB | BENIGN | SVEB | Q | F |
|---|---|---|---|---|---|---|
| DMTAE | 60% | 99.17 | 62.94 | 60.53 | 90.41 | **100.00** |
| MMD-AE | 60% | 98.58 | 63.63 | 44.90 | 93.33 | 89.66 |
| NSAE | 60% | 97.18 | 72.71 | 41.70 | 80.00 | 90.17 |
| CORAL | 60% | 97.87 | 58.24 | 69.70 | 95.26 | 73.33 |
| MTLS-RED | 60% | **99.34** | **77.49** | **73.48** | **95.52** | 93.33 |

Table 3: Performance (accuracy) of the proposed and baseline method with percentages of anomalies added to the source domain BENIGN class during training with Arrhythmia dataset. Here, we train the model on normal and VEB classes and use all the other anomaly classes as test OOD classes.

| MODEL | Domain1 | Domain2 | Domain3 | Domain4 |
|---|---|---|---|---|
| DIFEX | $65.02 \pm 2.00$ | $66.15 \pm 2.50$ | $64.06 \pm 2.00$ | $62.98 \pm 2.00$ |
| MTL-RED | $56.41 \pm 3.40$ | $56.30 \pm 2.50$ | $55.92 \pm 2.00$ | $55.82 \pm 2.50$ |

Table 4: Performance (accuracy %) of the proposed and DIFEX with EMG dataset divided into 4 domains each consisting of 6 classes for all the 9 persons.

on normal and VEB classes and use all the other anomaly classes SVEB, Q, and F as test OOD classes. In Table 4, we report the results of our model with the EMG dataset which is a time-series data. We randomly divide 36 subjects into four domains without overlapping and each domain contains data of 9 persons similar to DIFEX paper. While we find that the DIFEX model still shows better performance, we believe that this might be attributed to the fact that the dataset has just 8 features, unlike the high-dimensional datasets we have used for our experiments. Therefore, latent dimension analysis of such low-dimensional data might have resulted in a serious loss of information. Again, latent space anomaly detection is useful only because detecting anomalies is difficult only in the high dimensional space.

The overall experiment results show that although in some cases there is a decrease in performance in some of the source domains, an improvement in performance across all the OOD/unseen anomaly classes can be observed. In Figure 4, we plot the latent space of a simple multi-task encoder-decoder model without any regularization and our proposed model with the MI penalty. We see improved clustering with many of the source, cross, and target domain classes when the regularization is applied.

## 5 Conclusion

In this paper, we attempt to study an important problem of domain generalization in the context of anomaly detection where a model trained on several source and cross-domain anomaly classes can generalize better to completely unseen (OOD) classes of anomalies. We showcase that learning feature representation from multiple related domains in the latent space along with proper regularization designed to remove spurious correlations of the high-dimensional space can result in better generalization to unseen classes. We formulate a joint objective that aims to classify data in a multi-domain latent space while simultaneously enforcing a de-correlation penalty in the form of mutual information between input and latent space to create a domain-invariant disentangled latent space. We also show that cross-domain data can be usefully harnessed to enhance the generalization capability of classification models to unknown domains as it gives a way to add spurious correlation. Through detailed experiments, we showcase the performance of the model on benchmark IoT, healthcare, and time-series datasets in terms of accuracy. Since the kernel bandwidth is sensitive to the mutual information estimation, it would be interesting to learn this parameter during the training process. In the future, we would like to expand our research to domain generalization in multi-variate time series data with latent space regularization techniques.

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

# A  Intuition behind the Mutual Information Penalty

We consider a neural network encoder $E$, encoding input $X$ to latent space $Z$. Here, our goal is to remove the dependency of the encoder of the spurious correlation that is inherited from the high dimensional input space $X$ to the latent space $Z$. To achieve this, we augment the multi-task cross-entropy objective with mutual information between the original input space $X$ and latent space $Z$ which aims to create a disentangled or invariant latent space.

$$\mathcal{L}\left(\phi, \theta; X, Z\right) = \mathcal{L}_{ce} + \beta MI\left(X, Z\right) \tag{7}$$

where, $\mathcal{L}_{ce}$ is the generic classification loss. $MI(X, Z)$ is the mutual information loss between the encoding $Z$ and input space $X$. We aim to penalize large spurious information in the encoding space $Z$ and the penalty strength is determined by the parameter $\beta$. We consider the scenario where we do not know the specific features in the domains that have spurious correlation beforehand, therefore we directly minimize the MI content in the kernel space of the input and the latent representations which indirectly will aim to remove the spurious correlation in the high dimensional prior space and create a disentangled latent space. The mutual information(MI) content between the latent space and the input space can be expressed in the form of their joint and marginal distributions. By enforcing decorrelation, we minimize the MI divergence between joint and the product of the marginal distributions. Since MI is intractable to compute in general, the lower bound on the MI Phuong et al. (2018) can be given by:

$$MI\left(E_\phi\left(X\right), Z\right) = \mathbf{E}_{P_{X,Z}}\left[D_\theta\left(E_\phi\left(X\right), Z\right)\right] - \log \mathbf{E}_{P_X \bigotimes P_Z}\left[e^{D_\theta\left(E_\phi\left(X\right), Z\right)}\right]$$
$$\leq MI\left(E_\phi\left(X\right), Z\right)$$

The final objective can be written as,

$$\min_{\theta, \phi} \mathbf{E}\left[-\log p_\theta\left(Y|X\right)\right] + \beta MI\left(E_\phi\left(X\right), Z\right) \tag{8}$$

The classifier or the encoder tries to enforce de-correlation by minimizing the $MI(X, Z)$ in the kernel space of the input ($\mathcal{K}_x$) and the latent representations ($\mathcal{K}_z$) simultaneously with the cross-entropy loss. $\beta$ controls the trade-off between the de-correlation and the classification loss. The numerical stable objective is the sum of these two loss functions. The first part promotes discrimination power while the second part reduces the classification dependence on the spurious features in $X$.

## B    Differentiability of the Mutual Information Loss

Considering Renyi's second-order entropy where $\alpha = 2$, the entropy or the uncertainty of the input space can be estimated as

$$\hat{H}_2(\mathcal{K}_X) = -\log_2(tr(\mathcal{K}_X^2)) = -\log_2\left(\sum_{i=1}^{N} \lambda_i(\mathcal{K}_X)^2\right), \tag{9}$$

where, $\lambda_i(\mathcal{K}_X)$ denotes the $i^{th}$ eigenvalue of $\mathcal{K}_X$. We used this measure to estimate the entropy of the original input data space. Similarly, the entropy of the latent space can be estimated as

$$\hat{H}_2(\mathcal{K}_Z) = -\log_2(tr(\mathcal{K}_Z^2)) = -\log_2\left(\sum_{i=1}^{N} \lambda_i(\mathcal{K}_Z)^2\right), \tag{10}$$

. Now, the joint entropy between input and latent space can be measured as,

$$\hat{H}_2(\mathcal{K}_X, \mathcal{K}_Z) = H_2\left(\frac{\mathcal{K}_X \circ \mathcal{K}_Z}{tr(\mathcal{K}_X \circ \mathcal{K}_Z)}\right), \tag{11}$$

where $\circ$ denotes the Hadamard product of the matrices. Taking the analytical gradients, we can write:

$$\frac{\partial H_2(\mathcal{K}_X)}{\partial X} = -2\frac{\mathcal{K}_X}{tr(\mathcal{K}_X^2)} \tag{12}$$

Similarly, the joint entropy between prior and latent space can be estimated as

$$\frac{\partial H_2(\mathcal{K}_X, \mathcal{K}_Z)}{\partial X} = -2\left[\frac{(\mathcal{K}_X \circ \mathcal{K}_Z) \circ \mathcal{K}_Z}{tr(\mathcal{K}_X \circ \mathcal{K}_Z)^2} - \frac{MI(\mathcal{K}_X, \mathcal{K}_Z) \circ \mathcal{K}_Z}{tr(\mathcal{K}_X \circ \mathcal{K}_Z)}\right] \tag{13}$$

where, $MI(X, Z)$ is the mutual entropy between the prior and the latent space.

Now, the conditional entropy between latent and prior space can be estimated as:

$$\frac{\partial MI_2(\mathcal{K}_X; \mathcal{K}_Z)}{\partial X} = \frac{\partial H_2(\mathcal{K}_X)}{\partial X} + \frac{\partial H_2(\mathcal{K}_X, \mathcal{K}_Z)}{\partial X} \tag{14}$$

## C    Baselines

### C.1    DMTAE

This is a deep multi-task supervised autoencoder that optimizes the reconstruction loss of multiple source and cross-domains to improve the generalization performance on target domains. For $M$ source and cross-domains, and encoder and decoder parameters as $\phi$ and $\theta$ respectively, the reconstruction objective can be defined as

$$\mathcal{L}_{rec}(S_{train}; \theta, \phi) = \sum_{i=1}^{M} \left\| f_\theta^{(i)}\left(g_\phi^{(i)}\left(X^i\right)\right) - X^i \right\|_2^2 \tag{15}$$

The final supervised objective in the latent space for $N$ batches each having $l$ samples can be defined as

$$\mathcal{L}(S_{train}, Z; \phi, \theta) = \min_{\phi,\theta} \frac{1}{N} \sum_{l=1}^{N} \Big\{ \mathcal{L}_{ce}\left(g_\phi\left(X^l\right), y^l\right)$$
$$+ \lambda \mathcal{L}_{rec}\left(X^l; \phi, \theta\right) \Big\}, \tag{16}$$

where, $\mathcal{L}_{ce}$ is the cross-entropy loss calculated on the latent space encoding considering the binary classification problem. $Z$ and $X$ are the latent space and higher dimensional representations.

## C.2 MMD-AE

This paper uses the Maximum Mean Discrepancy measure for feature learning. Feature discrepancy is essentially a measure of distribution differentiation. MMD maps two distributions to Hilbert space, and judges the similarity between distributions by the sample mean discrepancy in the new space.

$$\mathcal{L}\left(\mathcal{S}_{train}, Z; \phi, \theta\right) = \min_{\phi,\theta} \frac{1}{N} \sum_{l=1}^{N} \Big\{ \mathcal{L}_{ce}\left(g_{\phi}\left(X^l\right), y^l\right) + \lambda_1 \mathcal{L}_{rec}\left(X^l; \phi, \theta\right) \\ + \lambda_2 \cdot MMD(\mathcal{K}_x, \mathcal{K}_z) \Big\}, \tag{17}$$

where,

$$\mathrm{MMD}(\mathcal{K}_x, \mathcal{K}_z) = \frac{1}{n^2} \sum_{i=1}^{n} \sum_{j=1}^{n} \mathcal{K}_{x_{ij}} - \frac{2}{nm} \sum_{i=1}^{n} \sum_{j=1}^{m} \mathcal{K}_{x_{ij}} + \frac{1}{m^2} \sum_{i=1}^{m} \sum_{j=1}^{m} \mathcal{K}_{z_{ij}} \tag{18}$$

where, $\mathcal{K}_x$ and $\mathcal{K}_z$ are the kernel matrices corresponding to samples from distributions $X$ and $Z$ respectively, $n$ and $m$ are the samples from each distributions. In our case, $n = m$ for each batch during training.

## C.3 Deep CORAL

We extend the CORAL model in the latent space. Here, we try to bound the covariances between the source $(C_S)$ and cross-domain$(C_D)$ latent representations using the Frobenious distance between the covariance matrices during each batch of training. The objective is defined as,

$$\mathcal{L}\left(\mathcal{S}_{train}, Z; \phi, \theta\right) = \min_{\phi,\theta} \frac{1}{N} \sum_{l=1}^{N} \Big\{ \mathcal{L}_{ce}\left(g_{\phi}\left(X^l\right), y^l\right) \\ + \lambda \mathcal{L}_{rec}\left(X^l; \phi, \theta\right) + \beta \cdot \left\| C_S - C_D \right\|_F^2 \Big\}, \tag{19}$$

## C.4 NSAE

NSAE trains the model by jointly reconstructing inputs and predicting the labels of inputs as well as their reconstructed pairs. The third term is the classification loss of reconstructed data.

$$\mathcal{L}\left(\mathcal{S}_{train}, Z; \phi, \theta\right) = \min_{\phi,\theta} \frac{1}{N} \sum_{l=1}^{N} \Big\{ \mathcal{L}_{ce}\left(g_{\phi}\left(X^l\right), y^l\right) \\ + \lambda_1 \mathcal{L}_{rec1}\left(X^l; \phi, \theta\right) + \lambda_2 \mathcal{L}_{rec2}\left(X^l; \phi, \theta\right) \Big\}, \tag{20}$$

We keep the NN architecture the same in all cases for a fair comparison.

## C.5 DIFEX

This paper proposes a custom loss function which encourages both internal invariance (within a domain) and cross domain feature learning to achieve better domain generalization.

$$\min_{\theta_f, \theta_c} \mathbb{E}_{(x,y) \sim p_{\mathrm{tr}}} \Big[ \mathcal{L}_{\mathrm{cls}}(G_c(G_f(x)), y) + \lambda_1 \mathcal{L}_{\mathrm{mse}}(z_1, G_f(\tilde{x})) + \lambda_2 \mathcal{L}_{\mathrm{align}} + \lambda_3 \mathcal{L}_{\mathrm{exp}}(z_1, z_2) \Big], \tag{21}$$

where,

- $\min_{\theta_f, \theta_c}$: Indicates that the optimization is with respect to the parameters $\theta_f$ (associated with a feature extractor or generator $G_f$) and $\theta_c$ ( associated with a classifier $G_c$).

- $\mathbb{E}_{(x,y) \sim p_{\mathrm{tr}}}$: The expectation is taken over the training data distribution $p_{\mathrm{tr}}$, where $x$ represents the input and $y$ represents the corresponding label.

- $\mathcal{L}_{\mathrm{cls}}(G_c(G_f(x)), y)$: This is the classification loss, where $G_f(x)$ represents the features extracted from the input $x$, and $G_c(G_f(x))$ represents the output of the classifier. This loss measures how well the model predicts the label $y$.

- $\lambda_1 \mathcal{L}_{\mathrm{mse}}(z_1, G_f(\tilde{x}))$: This term represents a Mean Squared Error (MSE) loss between a latent representation $z_1$ and the features extracted by $G_f$ from a transformed or perturbed input $\tilde{x}$. The hyperparameter $\lambda_1$ controls the importance of this loss term.

- $\lambda_2 \mathcal{L}_{\mathrm{align}}$: This term represents an alignment loss, between two different domains using the correlation alignment approach. These helps to cultivate the mutually invariant features. $\lambda_2$ controls the weight of this alignment loss.

- $\lambda_3 \mathcal{L}_{\mathrm{exp}}(z_1, z_2)$: This term regularize the distance between the internally-invariant ($z_1$) and mutually-invariant ($z_2$) features by maximizing their divergence, which they call the exploration loss. $\lambda_3$ controls the weight of this loss, which might be used to encourage disentanglement or decorrelation.

