# OpenReview forum: "Improving Out-of-Distribution Anomaly Detection with Domain-Invariant Latent Representations"
_TMLR — Rejected by TMLR_

### Review · Reviewer_aokR · 2024-07-13

**Summary Of Contributions:**

This paper addresses domain generalization by utilizing training data from multiple related domains to improve inference on unseen in-distribution (IN) and out-of-distribution (OOD) domains. It introduces a multi-task representation learning technique to enhance the detection of classes from unseen domains. The method creates a latent space that spans multiple domains, aiming to improve generalization to OOD domains. To achieve this, the authors attempt to minimize mutual information between the input and the latent space, reducing spurious correlations within a specific domain. This joint optimization promotes domain-invariant feature learning. The model's effectiveness is evaluated using standard classification metrics on cybersecurity datasets, comparing it to contemporary domain generalization methods.

**Audience:**

Yes

**Claims And Evidence:**

No

**Requested Changes:**

Please see weaknesses

**Strengths And Weaknesses:**

Strengths

- The problem considered in this paper is important.


Weaknesses


- The concepts of utilizing training data from multiple related domains, minimizing mutual information, and multi-task learning are quite common. Please clarify the main technical contributions of the paper.


- The paper missed several related works, and there are state-of-the-art methods for out-of-distribution (OOD) detection that the authors should consider, e.g.,

Xu, Ruijia, et al. "Deep cocktail network: Multi-source unsupervised domain adaptation with category shift." Proceedings of the IEEE conference on computer vision and pattern recognition. 2018.

Sun, Yiyou, et al. "Out-of-distribution detection with deep nearest neighbors." International Conference on Machine Learning. PMLR, 2022.

Yang, Jingkang, et al. "Openood: Benchmarking generalized out-of-distribution detection." Advances in Neural Information Processing Systems 35 (2022): 32598-32611.


- Experiments should be conducted on various types of datasets, rather than solely considering IoT datasets.


- Code should be provided to ensure reproducibility.

---

> ### Author Response · Authors · 2024-07-13
> **Main idea, datasets and baselines**
>
> Thanks for the reviews. Our main idea is to improve OOD class detection(those not used in the training phase) by incorporating data from both IN/SOURCE and CROSS domains with a custom loss function. In section 4.3.1, we explained how we select the IN/SOURCE and CROSS domains based on the feature correlation and kernel properties of those data. Equation 1 represents the multi-task learning objective which incorporates data from multiple IN and CROSS domains. This is the same objective as Ghifary et al. Our main objective is derived in eqn 6 where the third component is the multi-domain reconstruction loss. We derived the complete loss function stepwise for better understanding. The main intuition behind this custom loss function is to incorporate a regularization penalty that tries to remove the spurious correlation of the input space and create a disentangled latent space for better generalization. The latent space analysis makes sure the method works well with high-dimensional data as well. To achieve this, we use data from multiple domains(IN and CROSS) in a principled way to achieve this domain generalization. We agree with the reviewer that multi-task learning has been used for DG, but how we can employ a decorrelation penalty in the form of MI to directly disentangle a multi-domain latent space and improve generalization performance to OOD classes is the key idea. Experiment results show that we sometimes achieve good OOD accuracy at the expense of minor reduction in accuracy in IN and CROSS domain classes.
>
> We have used 3 standard IOT cybersecurity(CS) datasets, as latest as 2024 from the Canadian Institute of CS. CS datasets are preferred because of our funding liability. However, our method can be extended to any tabular form of iid data including medical, climate, and manufacturing, etc where rare/unseen anomaly detection is a crucial problem. To make our experiments more comprehensive, we will experiment on 2 standard healthcare multi-class datasets Arrhythmia and EMG, and report the results in the final version.
>
> For the baselines, we have selected those that have done latent space analysis and use supervised learning techniques, mostly with autoencoder or encoder/decoder-based methods. Therefore, our baselines consist of mainly regularization-based methods like CORAL, NSAE, MMD-AE, MTAE. Please check our ablation part where we show the latent dimension visualization of our method vs  without regularization(MTAE) and show better performance in detecting anomalies with most of the OOD, IN, and CROSS domain classes. The baselines that the reviewer mentioned use adversarial learning or distance-based metrics for OOD detection. As there are so many methods used for OOD domain generalization including meta-learning, metric learning, information bottleneck, representation learning with regularization etc, our method falls under the last category i.e representation learning with regularization.
>
> A closest paper to our work and published in TMLR is: Domain-invariant Feature Exploration for Domain Generalization(DG)
> https://openreview.net/forum?id=0xENE7HiYm. The authors of the paper also use a custom loss function to learn domain invariant features from multiple cross-domains and extend CORAL paper. However, they used the Fourier features for DG and the datasets we used are tabular iid data and not time series data. Hence, we did not include it in our baselines. If the reviewer has some specific method in mind, we will be happy to add it to our baselines.
>
> The link to the code is already mentioned in the paper.
> https://anonymous.4open.science/r/MTRAE-712B/mtrae/multitaskAE_orig.py

---

### Review · Reviewer_hPK3 · 2024-07-18

**Summary Of Contributions:**

The main contributions for this paper is to trying to propose a multi-task representation learning approach that adapts a multi-domain latent space by incorporating data from multiple sources and cross-domains.

The Multi-task Latent Space Regularized Encoder-Decoder Model (MTLS-RED) was proposed to jointly optimize several losses including classification loss, the multi-domain reconstruction loss, and the mutual invariance regularization in the latent space.

**Audience:**

Yes

**Broader Impact Concerns:**

I have not seen there is section inside the paper about explaining the broder impact concerns.

**Claims And Evidence:**

Yes

**Requested Changes:**

In this paper, what really needs to be changed is the detailed explanation of the nouns/tasks or this paper needs to be re-organized.

**Strengths And Weaknesses:**

Strengths: the proposed algorithms in regularizing the loss is understandable. The figures 2 representing the experiments are interesting.

Weakness:
1. General weakness
1.1 *Out-of-distribution detection* as mentioned in the title is a proper noun which refering to [1]. What the paper does is actually detection in OOD domain which is in-accurate.
1.2 There is no explain on why building the papre upon on the architecture of encoder-decoder which is not the normal way in doing OOD.
1.3 Why using different domains' information can enhance OOD detection remains un-explored.
1.4 What you trying to use is to conduct a multi-task representation learning, multi-domain is not the multi-task.

2. Details
2.1 The width of the figure 2 inside the paper is out of bound of the page.
2.2 There are two unexpected superlink for in page 6 and 7.

[1] Hendrycks, Dan and Kevin Gimpel. “A Baseline for Detecting Misclassified and Out-of-Distribution Examples in Neural Networks.” ArXiv abs/1610.02136 (2016): n. pag.

---

> ### Author Response · Authors · 2024-07-22
> **Reply to Reviewer hPK3**
>
> Thank you for the reviews on our work.
>
> 1.Title Clarification: We will surely consider changing the title to avoid confusion with the word "Out of Distribution detection". Our main idea is to improve OOD class detection(those not used in the training phase) by incorporating data from IN and CROSS domains with a custom loss function. In section 4.3.1, we explained how we select the IN and CROSS domains based on those data's feature correlation and kernel properties. In various applications like cybersecurity, healthcare, manufacturing etc, detecting the unseen classes/ "zero-day" attacks is crucial which necessitates a generalized model that can use the knowledge from existing/seen attack classes to detect those never seen classes.
>
> 2. Encoder-decoder modeling: In our related works, we briefly explained the different methodologies employed for OOD detection. There are many methods used for OOD domain generalization including meta-learning, metric learning, information bottleneck, representation learning with regularization etc, our method falls under representation learning with regularization. The latent space analysis ensures that we can extend the model for high-dimensional data as well. Moreover, by minimizing mutual information we aim to encourage the latent variables to be more compact or disentangled. This can lead to a more structured or simpler latent space but may also limit how much information the latent variables retain about the input. Therefore, just focusing on minimizing mutual information can conflict with the goal of achieving good reconstruction. This is exactly reflected in our experiments while choosing the values of the hyperparameters and we choose the reconstruction loss hyperparameter to be 3-4 times the MI loss to balance it during the training process. Increasing or decreasing these values beyond a particular range significantly impacts the classification performance on all the source, cross and OOD domains as shown in fig 3.
>
> Another key reason for choosing the latent space for detecting those unseen anomaly classes is the fact that the principle of concentration of distance (Beyer et al. 1999) reveals that for a query point p, its relative distance (or contrast) D to the farthest point and the nearest point converges to 0 with the increase of dimensionality d. This means that the discriminative power between the nearest and the farthest neighbor becomes rather poor in high-dimensional space.
>
> Beyer, K.; Goldstein, J.; Ramakrishnan, R.; and Shaft, U. 1999. When is “nearest neighbor” meaningful? In International Conference on Database Theory, 217–235. Springer
>
> 3. Using different domain data: Adding cross-domain data can help improve generalization performance if it obeys certain kernel properties of the data. We consider a setting where joint distribution of the data doesn’t have much overlap but the marginal distribution of the features does have an overlap Dong & Ma (2022). The authors claim that two datasets must have overlapping marginal distribution which is required for extrapolation. We combine the datasets with different patterns of correlation structure among its features during the training phase. We find that this gives better generalization results than randomly combining any two attack classes. In all our cases, we keep the rare anomaly classes(with the least nos of samples available) as the OOD class. Incorporating data with varying correlation structures effectively introduces noise during training, which serves as a straightforward and natural method to bias the network towards learning invariant representations.
>
> Kefan Dong and Tengyu Ma. First steps toward understanding the extrapolation of nonlinear models to unseen domains.
>
> 4. Multi-task representation learning/ multi-domain - Ours is a multi-task representation learning model with a regularization penalty to remove spurious correlation in latent space. Since we used data from multiple classes(each dataset is an anomaly class), we referred to it as "multi-domain". We would like to refer to a related paper at TMLR "Domain-invariant Feature Exploration for Domain Generalization(DG)", where the authors also used related domains in their training process. For e.g, they used the  VLCS dataset which comprises photographic domains (Caltech101, LabelMe, SUN09, VOC2007). For each dataset, they leave one domain as the test domain which is unseen in training while the others as the training domains. Similarly, in our case, we have trained our model on related cybersecurity/IoT datasets where each of these classes is a different kind of cyber attack dataset. For each case, we kept 2-3 domains as OOD classes which are not used in the training process and are used for evaluation purpose only. We are a little confused about why the reviewer mentioned that we cannot refer to the different attack datasets as "multiple domains". Kindly clarify.
>
> 5. We will adjust the figures.

---

### Review · Reviewer_tiWw · 2024-08-12

**Summary Of Contributions:**

This paper proposes a domain generalization method that introduces two regularizations: minimizing the mutual information between input and latent features, and incorporating a reconstruction loss. Experiments conducted on various cybersecurity and IoT datasets demonstrate that the proposed methods perform better than some previous approaches in these datasets.

**Audience:**

Yes

**Claims And Evidence:**

Yes

**Requested Changes:**

See the comments.

**Strengths And Weaknesses:**

I would like to offer some constructive feedback on this paper, as there are a few areas that could benefit from improvement.

1. The title of the paper references "out-of-distribution detection," but it's not entirely clear how "detection" is being addressed. The focus seems to be more on domain generalization, specifically on multi-domain problems. However, the definition of the tasks, particularly the role of "multi-task," is unclear. Additionally, the method for calculating "accuracy" in the experimental results needs further clarification, especially considering that the unseen domain may have labels that are not present in the source domain.

2. The paper makes some ambitious claims about decorrelating spurious correlations and disentangled representations. It’s important to note that minimizing the mutual information between the input and latent space does not necessarily imply that spurious correlations are decorrelated, nor does it ensure disentanglement. These are distinct concepts and lack a causal relationship.

3. When using a deep network as an encoder, the input and latent features form deterministic mappings. As a result, the mutual information between the input and latent features is actually equivalent to the entropy of the latent features. It would be beneficial to ensure that the regularization is being computed correctly in light of this.

4. While minimizing mutual information is a widely used tool in the community, applying it to domain generalization is not a novel idea. The discussion of related work on this topic could be more comprehensive. Additionally, the comparison methods include some outdated approaches, and there is a lack of discussion about more recent techniques.

5. The minimization of mutual information and reconstruction are actually conflicting objectives; reconstruction implies maximizing mutual information. Combining them in this paper does not seem very reasonable.

---

> ### Author Response · Authors · 2024-08-13
> **Reply to Reviewer tiWw**
>
> Thank you for the reviews on our work.
> 1. Title and role of multi-task: We will surely consider changing the title to avoid confusion with the word "Out of Distribution detection". Our main idea is to improve OOD class anomaly detection(those not used in the training phase/unseen) by incorporating data from IN and CROSS domains with a custom loss function. In section 4.3.1, we explained how we select the Source and CROSS domains based on those datasets' feature correlation and kernel properties.  Multi-task learning allows combining multiple related domains and learning their feature representations in a latent space to improve the classification of unseen classes. The reconstruction loss in eq 1 enables combining features from multiple related domains/classes. Accuracy is calculated on all the source, cross, and OOD domains.
> 2.  "minimizing the mutual information between the input and latent space does not necessarily imply that spurious correlations are decorrelated" - We humbly disagree with the reviewer here. We will refer to a significant work in this domain to establish our point - "Measuring Dependence with Matrix-based Entropy Functional". The authors develop the matrix-based normalized total correlation(TC) to measure non-linear dependence between more than two variables. We use Renyi's 2-order entropy to measure the entropy or the uncertainty of the data in the eigen spectrum of a Gram matrix in the kernel space. We try to minimize the mutual information between the input and the latent space to develop the disentangled latent space.  Using the sufficiency and minimality conditions of the information bottleneck principle we try to create a latent space representation that contains only relevant information about the output tasks and discard the class-specific spurious correlation information of the input space. Therefore, the latent space has now more relevant information. We show that the matrix-based dependence measure can be applied for domain generalization in a multi-task learning scenario by using it as a regularization method to remove the spurious class-specific correlation information of input space.  In particular TC(Z) is zero if and only if the components of Z are independent, in which case we say that Z is disentangled.
>
> $MI(Z;X) = \mathbb{E}_{x \sim p(x)} \text{KL}(p(Z|X) \parallel p(Z))$
>
> We say that $\(X, Z, Y\)$ form a Markov chain, Y is the downstream task, indicated with $\(X \rightarrow Z \rightarrow Y\)$, if $\(p(Y|X, Z) = p(Y|Z)\)$.
>
> Yu, Shujian, et al. "Measuring dependence with matrix-based entropy functional." Proceedings of the AAAI Conference on Artificial Intelligence, 2021
>
> 3. "mutual information between the input and latent features is actually equivalent to the entropy of the latent features" - It is true that unless it is probabilistic, the input and latent space form deterministic mappings. However, the quality and utility of the deterministic mapping depend on how well the latent space captures the relevant features of the input data. In our case, we train the network to ensure that this mapping is useful for downstream tasks which in our case is to detect unseen classes of anomalies.
> 4. "minimizing mutual information is a widely used tool in the community"- Our idea is not just to minimize MI but we design a custom loss function that considers data from multiple domains and uses the regularization penalty to create a disentangled latent space to improve OOD anomaly detection. In our case, incorporating data with varying correlation structures effectively introduces spurious signal /noise during training, which serves as a straightforward and natural method to bias the network toward learning invariant representations
> 5. "minimization of mutual information and reconstruction are actually conflicting objectives;"  - When minimizing mutual information in our regularization approach, the aim is to encourage the latent variables to be more compact or disentangled, potentially reducing redundancy. This can lead to a more structured or simpler latent space but may also limit how much information the latent variables retain about the input. Therefore, just focusing on minimizing mutual information can conflict to achieve good reconstruction. This is exactly reflected in our experiments while choosing the values of the hyperparameters $\lambda$ and $\beta$. Increasing or decreasing these values beyond a particular range significantly impacts the classification performance on all the source, cross and OOD domains as shown in fig 3. A simple example of conflicting loss is a VAE -  the reconstruction loss measures how well the model can reconstruct the original input from the latent representation by retaining enough information to accurately regenerate the input data whereas the KL divergence loss regularizes the latent space by encouraging the learned latent distribution to be close to a prior distribution..

---

> ### Author Response · Authors · 2024-08-14
> **Reply to Reviewer tiWw**
>
> 1. Another important reason for choosing the latent space for detecting those unseen anomaly classes is the fact that the principle of concentration of distance (Beyer et al. 1999) reveals that for a query point p, its relative distance (or contrast) D to the farthest point and the nearest point converges to 0 with the increase of dimensionality d. This means that the discriminative power between the nearest and the farthest neighbor becomes poor in high-dimensional space. Therefore anomaly detection problems usually consider the latent space embeddings instead of high-dimensional features.
>
> Beyer, K.; Goldstein, J.; Ramakrishnan, R.; and Shaft, U.
> 1999. When is “nearest neighbor” meaningful? In International Conference on Database Theory, 217–235. Springer
>
> 2. In eq. 5, we calculate the renyi's entropy of the input space X (samples * features), latent space Z, and the joint entropy between X and Z to calculate the MI between input space X and latent space Z. Minimizing this MI indirectly minimizes the total correlation between input and latent space features. Minimizing the feature correlation(de-correlate) between the input and latent spaces is the way we try to cultivate disentanglement in the latent space representations. Definition 2 of the mentioned paper provides the details of how it is measured. Since this is not our main contribution, we just cited the paper and did not provide the details of its theory.
>
> Yu, Shujian, et al. "Measuring dependence with matrix-based entropy functional." Proceedings of the AAAI Conference on Artificial Intelligence, 2021.
>
> 3. " introduces two regularizations: minimizing the mutual information between input and latent features, and incorporating a reconstruction loss" - we would like to clarify here that the reconstruction loss is an integral part of a latent space model and the only regularization we introduced is the MI penalty between latent and input space. We refer to Fig: 4, where our comparison is with a supervised encoder-decoder model without the regularization. We see improvement in the latent space separation of normal and anomaly samples for most attack classes.
>
> 4. Another theoretical paper in this direction claims " invariance to nuisance factors in a deep neural network is equivalent to information minimality of the learned representation, and that stacking layers and injecting noise during training naturally bias the network towards learning invariant representations." In our case, the cross-domain data with different kinds of correlation structures helps adding spurious sginals/noise. The MI minimization aims to remove that noise to create a disentangled latent space.
>
> Achille, A. and Soatto, S., 2018. Emergence of invariance and disentanglement in deep representations. Journal of Machine Learning Research, 19(50), pp.1-34.

---

### Comment · Action_Editor_Np3q · 2024-09-10
**Official Recommendation**

Dear Reviewers,

Please review the authors’ rebuttal, the other reviewers’ comments, and the revised manuscript. Once completed, submit your final recommendation. Thank you again for your time and effort in reviewing this submission.

Kind regards,

Your Editor

---

### Decision · Action_Editor_Np3q · 2024-09-24

**Recommendation:** Reject

**Comment:**

Thanks for providing the response and the revised version. While some of the problems have been addressed, this submission still has several major issues.

- First, anomaly detection, out-of-distribution (OOD) detection, and OOD generalization are distinct tasks with different goals. The change of title to anomaly detection remains confusing. As reviewer hPK3 comments, “The core idea focuses on Out-of-Distribution generalization instead of anomaly detection… which makes the paper’s content hard to understand.” After reviewing the revision, I agree with this observation and suggest another round of major revisions to present the method clearly and accurately.

- Second, all reviewers have pointed out that the techniques used (minimizing mutual information and multi-task learning) are common in OOD generalization. While it is good to see their application to a new dataset/task, there should be more modifications or insights to convince the reviewers of the contribution of this submission. The clarification provided—“Ours is a multi-task representation learning model with a regularization penalty to remove spurious correlations in latent space”—does not sufficiently address this concern or highlight the unique contribution of the work.

- Third, the current experiments are not solid enough to support the effectiveness of the proposed method. Reviewer tiWw mentions that “the comparison methods include some outdated approaches, and there is a lack of discussion about more recent techniques.” I share this concern. The revised version did not fully address this issue. Reviewer aokR comments, “Experiments should be conducted on various types of datasets, rather than solely considering IoT datasets.” This feedback suggests a broader evaluation is necessary. The rebuttal frequently cites “A closely related paper published in TMLR: Domain-invariant Feature Exploration for Domain Generalization (DG).” A more effective approach would be to use commonly employed datasets for comparison and include evaluations against recent methods to substantiate the method’s effectiveness.

Despite these issues, the authors are encouraged to continue refining their work. Applying existing techniques to new tasks or datasets is always welcome. It would be also beneficial to clearly outline the new issues in the new tasks/datasets and the strategies employed to resolve them in future revisions.

Based on the above points and the final comments of the three reviewers, the recommendation is to reject the submission.

**Audience:**

This submission would be of interest to researchers working in the areas of domain adaptation and generalization. Additionally, users looking to develop robust and effective models would likely find this paper valuable.

**Claims And Evidence:**

This work focuses on domain generalization, aiming to learn domain-invariant representations from samples across multiple domains. To achieve this, the submission proposes disentangling the latent space by minimizing the mutual information between inputs and outputs. It also addresses the decorrelation of spurious features within each domain.

In the rebuttal phase, the authors worked diligently to clarify the mechanism of the proposed module. However, Reviewer tiWw remains concerned about the effectiveness of the method: deterministic mapping between inputs and latent variables can make it difficult to properly define the random variables. This could lead to an ill-posed problem if not carefully managed.